# Multigene Panel Next-Generation Sequencing Techniques in the Management of Patients with Metastatic Colorectal Carcinoma: The Way Forward for Personalized Treatment? A Single-Center Experience

**DOI:** 10.3390/ijms252011071

**Published:** 2024-10-15

**Authors:** Laura Matteucci, Francesco Giulio Sullo, Chiara Gallio, Luca Esposito, Margherita Muratore, Ilario Giovanni Rapposelli, Daniele Calistri, Elisabetta Petracci, Claudia Rengucci, Laura Capelli, Elisa Chiadini, Paola Ulivi, Alessandro Passardi, Alessandro Bittoni

**Affiliations:** 1Department of Medical Oncology, IRCCS Istituto Romagnolo per lo Studio dei Tumori (IRST) “Dino Amadori”, Via P. Maroncelli 40, 47014 Meldola, Italy; laura.matteucci@irst.emr.it (L.M.); francesco.sullo@irst.emr.it (F.G.S.); chiara.gallio@irst.emr.it (C.G.); luca.esposito@irst.emr.it (L.E.); margherita.muratore@irst.emr.it (M.M.); ilario.rapposelli@irst.emr.it (I.G.R.); alessandro.bittoni@irst.emr.it (A.B.); 2Biosciences Laboratory, IRCCS Istituto Romagnolo per lo Studio dei Tumori (IRST) “Dino Amadori”, Via P. Maroncelli 40, 47014 Meldola, Italy; daniele.calistri@irst.emr.it (D.C.); claudia.rengucci@irst.emr.it (C.R.); laura.capelli@irst.emr.it (L.C.); elisa.chiadini@irst.emr.it (E.C.); paola.ulivi@irst.emr.it (P.U.); 3Unit of Biostatistics and Clinical Trials, IRCCS Istituto Romagnolo per lo Studio dei Tumori (IRST) “Dino Amadori”, Via P. Maroncelli 40, 47014 Meldola, Italy; elisabetta.petracci@irst.emr.it

**Keywords:** metastatic colorectal cancer, multigene panel, next-generation sequencing, molecular biomarkers, druggable targets

## Abstract

The efficacy and cost-effectiveness of Multigene Panel Next-Generation Sequencing (NGS) in directing patients towards genomically matched therapies remain uncertain. This study investigated metastatic colorectal cancer (mCRC) patients who underwent NGS analysis on formalin-fixed paraffin-embedded tumor samples. Data from 179 patients were analyzed, revealing no mutations in 39 patients (21.8%), one mutation in 83 patients (46.4%), and two or more mutations in 57 patients (31.8%). *KRAS* mutations were found in 87 patients (48.6%), including *KRAS* G12C mutations in 5 patients (2.8%), *PIK3CA* mutations in 40 patients (22.4%), and *BRAF* mutations in 26 patients (14.5%). Less common mutations were identified: *ERBB2* in five patients (2.8%) and *SMO* in four patients (2.2%). Additionally, *MAP2K1*, *CTNNB1*, and *MYC* were mutated in three patients (2.4%). Two mutations (1.1%) were observed in *ERBB3*, *RAF1*, *MTOR*, *JAK1*, and *FGFR2*. No significant survival differences were observed based on number of mutations. In total, 40% of patients had druggable molecular alterations, but only 1.1% received genomically guided treatment, suggesting limited application in standard practice. Despite this, expanded gene panel testing can identify actionable mutations, aiding personalized treatment strategies in metastatic CRC, although current eligibility for biomarker-guided trials remains limited.

## 1. Introduction

Colorectal cancer (CRC) is the third leading cause of cancer-related death and accounts for 10% of cancer diagnoses worldwide [1]. In recent decades, personalized decision making has been empowered by the increasing use of comprehensive genomic profiling, which has allowed for the detection of novel actionable targets. As is known, EGFR inhibition with specific monoclonal antibodies has become the standard of care in *RAS* wild-type patients, providing significant benefits in terms of survival when administered in combination with chemotherapy [2,3]. More recently, specific *RAS* mutations such as *KRAS* G12C have gained interest as possible actionable targets, following the evaluation of sotorasib or adagrasib in combination with anti-EGFR [4,5]. The actionability of other mutations such as *BRAF* V600E has also provided encouraging results, leading to the approval of the BRAF inhibitor encorafenib in combination with anti-EGFR cetuximab [6]. Larger aberrations such as *HER-2* amplification have also been investigated in several studies evaluating the role of different combination strategies, which provided interesting results [7,8,9]. Genomic profiling has thus widened the application of targeted therapy in digestive oncology [10,11], and also empowered the use of immune checkpoint inhibitors (ICIs), whose efficacy is restricted to patients harboring microsatellite instability (MSI-H) and *POLE-D1* mutations [12,13]. Despite the encouraging results obtained in terms of survival and disease control, the employment of targeted agents is burdened by primary and secondary resistance [11,12]. For this reason, their use remains restricted to a limited proportion of patients.

The advent of high-throughput techniques such as DNA and RNA sequencing has allowed for the global and unbiased evaluation of the genomic landscape of tumors, enabling the identification of hotspot mutations, copy number variants, and fusion transcripts.

Nevertheless, the incidence of druggable molecular alterations is currently quite low, and the execution and interpretation of the Multigene Panel Next-Generation Sequencing genomic profile (NGS) may be expensive and require specific expertise which is not available in all centers. Therefore, the clinical utility and cost-effectiveness of the implementation of NGS in mCRC is still uncertain.

In our study, we analyzed a cohort of mCRC patients who underwent molecular characterization using NGS multigene technology with the Oncomine Focus Assay. The primary objective of this study was to evaluate the utility of Next-Generation Sequencing (NGS) in identifying actionable mutations in metastatic colorectal cancer (mCRC) patients and to assess its impact on clinical trial enrollment and personalized treatment strategies. Specifically, the study aimed to determine whether systematic NGS profiling could increase the number of patients eligible for genomically guided therapies, improving treatment outcomes.

## 2. Results

### 2.1. Patient Characteristics

Patients with a histologically confirmed diagnosis of colorectal adenocarcinoma and an NGS assay performed in our institution from June 2019 to December 2020 were included. A total of 179 patients were considered for the analysis.

Patient characteristics are summarized in Table 1. Eastern Cooperative Oncology Group Performance status (ECOG PS) at metastatic disease diagnosis was recorded in 160 patients (89.3%), with 140 patients (78.2%) showing an ECOG PS of 0 or 1, and 20 patients (11.1%) with an ECOG PS ≥ 2. Synchronous metastases were observed in 63.1% of cases (*n* = 113), whereas metachronous lesions were identified in 36.9% of cases (*n* = 66). Surgical resection of metastases was performed in 28% of cases, with hepatic resections being the most common (30%), followed by peritoneal (20%) and pulmonary metastasectomies (18%). In total, 149 patients (83.2%) underwent first-line chemotherapy, with 51.9% of patients (*n* = 93) receiving at least two lines of treatment afterwards. A total of 44.1% of patients (*n* = 66) received doublet or triplet chemotherapy combined with bevacizumab, and 24.9% (37 patients) received doublet or triplet chemotherapy with an EGFR inhibitor (cetuximab or panitumumab). Eight patients with NGS-detected actionable alterations were administered targeted therapies. Among these patients, six harbored the *BRAF* V600E mutation and received a combination of encorafenib and cetuximab, while two patients with *PI3K* alterations were treated with alpelisib, a PI3K inhibitor.

### 2.2. Molecular Alterations

No mutations were detected in 39 patients (21.8%) whereas 140 patients reported at least one mutation (78.2%), Figure 1. Table 2 shows the distribution of molecular alterations observed in the study population. The most frequent mutations were those in the *KRAS, PIK3CA*, and *BRAF* V600E genes. Some mutations were observed only in three patients such as those in the *CTNNB1*, *MYC*, and *MAP2K1* genes. Other mutations were even more rare, such as those in the *ERBB3*, *RAF1*, *MTOR*, *JAK1*, and *FGFR2* genes, detected each in only two patients, or those in the in *CDK4*, *MET*, *FGFR3*, *GNA11*, *EGFR*, *ALK*, *ROS1*, *DDR2*, and *KIT* genes, found in only one patient each. It is worth noting that no amplifications or fusions were identified and hotspot mutations were the only type of aberration detected (Single-Nucleotide Variants, SNVs, and insertions–deletions, INDELs). No mutations were observed for the 28 genes of the Oncomine Focus panel.

### 2.3. Association between Mutated Genes and Survival (PFS, OS)

Overall, the median follow-up time was 33 months (95% CI 28.45–not reached—NR), with a median PFS of 10.3 months (95% CI 8.8–12.3) and a median OS of 32.7 months (95% CI 24.8–39.2). The study investigated the association between the most frequent mutations, that is those occurring in at least 5% of patients, and PFS during first-line chemotherapy. Patients with a *KRAS* mutation exhibited a median PFS of 10.4 months (95% CI 8.84–12.25), compared to 10.1 months (95% CI 7.29–13.70) in *KRAS* wild-type patients (*p* = 0.737; HR 0.95, 95% CI 0.69–1.29). Patients harboring the *PIK3CA* mutation showed a median PFS of 10.7 months (95% CI 7.75–13.70), versus 10.0 months (95% CI 8.57–12.32) in *PIK3CA* wild-type cases (*p* = 0.942; HR 1.01, 95% CI 0.70–1.47). *BRAF*-mutated tumors showed a median PFS of 5.2 months (95% CI 3.52–9.86) compared to 10.7 months (95% CI 9.00–12.94) in *BRAF* wild-type tumors(*p* = 0.060; HR 1.54, 95% CI 0.98–2.40) (Figure 2, left panel), and *APC*-mutated tumors demonstrated a median PFS of 9.0 months (95% CI 1.81–12.32) versus 10.3 months (95% CI 8.84–12.52) in *APC* wild-type tumors(*p* = 0.330; HR 1.33, 95% CI 0.75–2.35).

The study also investigated the potential association between OS and the most frequently mutated genes. Analogous analyses for OS showed that patients with mutated *KRAS* exhibited a median survival of 34.9 months (95% CI 24.77–65.60) compared to 26.9 months (95% CI 22.37–39.19) in wild-type *KRAS* patients (*p* = 0.084; HR 0.71, 95% CI 0.48–1.05). Similarly, mutated *PIK3CA* patients displayed a median OS of 35.8 months (95% CI 23.85–NR) versus 30.2 months (95% CI 24.01–39.19) in wild-type *PIK3CA* patients (*p* = 0.270; HR 0.76, 95% CI 0.47–1.24). In contrast, mutated *BRAF* patients showed a median OS of 13.9 months (95% CI 4.89–26.05) versus 35.8 months (95% CI 26.91–48.59) in wild-type *BRAF* patients (*p* < 0.001; HR 2.62, 95% CI 1.59–4.32) (Figure 2, right panel). Additionally, mutated *APC* patients demonstrated a median OS of 24.8 months (95% CI 2.3–NR) versus 32.8 months (95% CI 24.8–41.8) in wild-type *APC* patients (*p* = 0.354; HR 1.36, 95% CI 0.71–2.62).

Among the less mutated genes, no differences were observed except for the *CTNNB1* gene, which was mutated in only three patients. Patients harboring a mutation in such a gene showed a statistically significant shorter PFS and OS compared to wild-type patients. In detail, patients with mutated *CTNNB1* had a median PFS of 1.8 months (95% CI 1.15–NR) compared to 10.4 months (95% CI 9.0–12.32) in *CTNBB1* wild-type patients (*p* = 0.001; HR 6.85, 95% CI 2.13–22.03). Similarly, the median OS in patients with mutated *CTNNB1* was 2.1 months (95% CI 1.87–NR) versus 32.8 months (95% CI 24.8–41.8) in *CTNBB1* wild-type patients (*p* < 0.001; HR 9.82, 95% CI 3.00–32.12).

No significant differences in terms of OS and PFS were found when comparing patients with high KRAS or BRAF allele frequency, defined as a Variant Allele Frequency (VAF) of more than the median value, versus low allele frequency.

No significant relationship was found between the number of mutated genes and survival.

### 2.4. Association between RAS Mutation Allele Frequency and Survival

We also investigated the association between *RAS* mutation (*KRAS* and *NRAS*) allele frequency and survival outcomes in patients treated with first-line chemotherapy (*n* = 148). Patients with a high *RAS* allele frequency, defined as Variant Allele Frequency (VAF) > 20%, showed a trend for lower survival rates with a median OS of 30.95 months (95% CI 18.2–47.4) compared to 37.42 months (95% CI 26.97–NR) for patients with low VAF (*p* = 0.091; HR = 2.13, 95% CI 0.89–5.09).

## 3. Discussion

To the best of our knowledge, there are currently few real-world evidence studies based on routinely collected data from standard clinical practice that investigate the impact of NGS on a large cohort of patients diagnosed with mCRC. One of the main objectives of this study was to assess the clinical value of genomic profiling using NGS techniques in a sample of treated patients referred to our oncology units.

No new genomic prognostic factors were identified with this NGS panel, and the *BRAF* V600E mutation was confirmed to be a negative prognostic factor, as previously reported in the literature [14,15,16]. While no statistically significant correlation was identified between the number of mutated genes and survival, there appears to be a negative survival trend in patients with at least one mutation. A statistically significant decrease in both PFS and OS was noted among patients harboring a mutation in the CTNNB1 gene. Nonetheless, due to the limited number of patients exhibiting this mutation, the reliability of the data, despite their statistical significance, has to be carefully interpreted, and no conclusive inferences can be drawn concerning their prognostic role.

A retrospective study by Nindra U et al. [17] assessed the clinical utility of a 50-gene NGS panel (Oncomine Precision Assay™) on 180 mCRC samples from an Australian population. The authors found at least one gene mutation in 147 (82%) patients while two or more concurrent mutations were identified in 68 (38%) patients. In this cohort, patients with concurrent TP53 and RAS mutations had significantly reduced overall survival as well as patients with high KRAS allele frequency (>20%). Interestingly, the NGS panel revealed 22% of cases with Tier II and III ESCAT mutations, according to the European Society of Medical Oncology (ESMO) Precision Medicine Working Group classification of actionability [18], suggesting that extended genomic profiling may increase the number of mCRC patients treated with targeted therapy.

The choice of the first-line therapeutic strategy was not influenced by NGS results in our cohort of patients. Indeed, in daily practice, the choice of the first-line treatment is based only on the mutational status of *KRAS* and *BRAF* as well as microsatellite instability. Therefore, NGS data were taken into consideration only at progression following the discontinuation of standard treatments, in order to allow enrollment in a clinical trial or the use of off-label or expanded-access therapies. Unfortunately, many of these heavily pretreated patients were unfit for subsequent lines of treatment. This may explain the low number of patients treated with molecular-targeted therapies. In a similar study on a cohort of 187 heavily pretreated mCRC patients, the use of NGS with a custom amplicon-based NGS assay (MiSeq) covering 60 genes yielded a final rate of inclusion into genomically guided clinical trials of 2.7%, only slightly higher compared to our study [19].

Molecular-targeted therapy in mCRC treatment is increasingly becoming a therapeutic option due to the availability of selective inhibitors. Therefore, centralizing NGS genomic profiling in high-volume reference laboratories and establishing teams of professionals capable of evaluating and interpreting complex molecular test results (Molecular Tumor Board) is increasingly necessary to identify molecular alterations that provide sensitivity to specific molecular-targeted therapies in subpopulations of otherwise difficult-to-treat patients.

This study has some limitations that are worth noting. First, molecular analysis was performed on the primary tumor or metastasis without taking into account, as per clinical practice, the spatial and temporal heterogeneity of colorectal cancer. One limitation of this study is also related to the NGS panel employed which was restricted to 52 genes. Broader panels with the ability to analyze hundreds of genes can indeed identify a greater number of druggable molecular alterations, define the tumor mutational burden, and detect increasingly informative mutations such as *POLE/D1* mutations which are predictive of sensitivity to immunotherapies.

## 4. Materials and Methods

### 4.1. Patient Selection and Data Collection

Patients aged 18 years and older diagnosed with mCRC who were NGS-tested with the Oncomine Focus Assay between June 2019 and December 2020 at the Biosciences Laboratory of IRCCS Istituto Romagnolo per lo studio dei Tumori (IRST) were included. Patients with other concurrent neoplasms or localized colorectal carcinoma were excluded.

For each included patient, demographic and clinical data were collected. Additionally, molecular information such as the quantity and type of molecular alterations detected as well as the variations of their allelic frequency (VAF) were recorded.

All participants provided written informed consent, and the study was conducted in compliance with the Declaration of Helsinki under good clinical practice conditions and with approval from the Local Ethics Committee (Comitato Etico Area Vasta Romagna e IRST). Information was extracted from patients’ electronic medical records in accordance with stringent privacy standards, and the data were anonymized and recorded in an Excel database.

### 4.2. Molecular Analyses

Formalin-fixed paraffin-embedded (FFPE) tumor samples were used for molecular analysis. For mutation identification, a “tumor-only” approach was employed. To mitigate the impact of FFPE-associated artifacts, we implemented rigorous quality control measures, ensuring the reliability and accuracy of the molecular data. The depth of coverage (>500×) in this study provided sufficient sensitivity to detect lower frequency variants, assuming high sequencing read quality. The following sequencing quality control (QC) metrics were applied: the number of mapped reads (DNA + RNA) ranged from 45 M to 75 M, the percent of reads on target was >90%, the average base coverage depth was >800×, and the uniformity of amplicon coverage was >90%, with more than 3781 amplicons and >90% assigned amplicon reads. For each sample, tumoral areas were defined by the pathologists, macrodissected, and collected in specific tubes. Nucleic acids were extracted using a MagMAX FFPE DNA/RNA Ultra Kit (Applied Biosystems, Waltham, MA, USA) following the manufacturer’s protocol. DNA and RNA concentrations were determined by fluorometric quantitation using a Qubit 4.0 Fluorometer with a Qubit DNA dsDNA HS Assay Kit and Qubit RNA HS Assay Kit (Thermo Fisher Scientific, Waltham, MA, USA), as appropriate.

NGS analyses were performed using an Oncomine™ Focus Assay panel (Thermo Fisher Scientific, Waltham, MA, USA), an amplicon-based DNA/RNA NGS assay that covers 52 cancer-associated genes. The DNA panel could identify hotspot mutations in 35 genes and copy number variants in 19 genes. The RNA panel was able to detect fusion drivers in 23 genes.

DNA and RNA libraries were prepared automatically using the library preparer “Ion Chef^TM^ System” (Thermo Fisher Scientific, Waltham, MA, USA) following the manufacturer’s instructions, with 10 ng of input DNA and RNA per sample.

Prior to RNA library preparation, complementary DNA (cDNA) synthesis was carried out using a SuperScript™ VILO™ cDNA Synthesis Kit (Thermo Fisher Scientific). The libraries were loaded onto an Ion Chef System (Thermo Fisher Scientific) for template preparation and finally sequenced on the Ion S5 Plus platform (Thermo Fisher Scientific) using Ion 520 Chips (Thermo Fisher Scientific). The primary analysis was carried out using a Torrent Suite Server™ 5.12.3 to perform initial quality control, including chip loading density, median read length, and number of mapped reads. Afterwards, a second analysis was performed by Ion Reporter™ Software 5.20, hosting informatics tools for variants, filtering, and annotations. The criteria for variant calling in this study were as follows: Only variants with a Variant Allele Frequency (VAF) greater than 5% (VAF > 5%) were considered, with a minimum coverage requirement of 500×. Benign, likely benign, and germ-line variants, including BRCA1/2, PMS2, and MLH1, were excluded from the analysis. For RNA analysis, only samples with total mapped reads greater than 500 K were included. Target fusions were considered when supported by at least 40 reads, while non-target fusions required a minimum of 1000 supporting reads. Copy Number Variants (CNVs) were included if detected in more than 50% of cancer cells and with a copy number greater than 7. Additionally, the MAPD (median absolute pairwise difference) had to be less than 0.3, and the *p*-value for significance was set at <0.05.

### 4.3. Statistical Analysis

A dedicated database was created to retrospectively collect demographic, clinical, histological, and treatment data. The patients’ characteristics were summarized using the median and first and third quartiles for continuous variables, and frequencies and percentages for categorical variables. PFS was defined as the time in months from the date of metastatic tumor diagnosis to progression or death from any cause, whichever occurred first, during the course of first-line chemotherapy. For patients alive without progression, the date of the last follow-up was recorded. OS was determined as the time in months from the date of metastatic tumor diagnosis to death from any cause or the date of the last follow-up. The median follow-up time was computed using the reverse Kaplan–Meier method whereas the survival curves were estimated using the Kaplan–Meier method, and comparisons were made using the log-rank test. Univariate Cox regression models were employed to estimate hazard ratios (HRs) and their corresponding 95% confidence intervals (CIs). With regard to the analysis of the association between the mutational status together with the variant allelic frequency (VAF) and survival, the cut-offs were based on the median value of the VAF distribution of a specific gene excluding the wild-type patients. Results were considered statistically significant if the two-sided *p*-values were <0.05. Statistical analyses were conducted using STATA 15.0 (College Station, TX, USA).

## 5. Conclusions

At present, molecular testing for MMR status as well as *KRAS*, *NRAS*, and *BRAF* mutations is recommended for all mCRC patients at diagnosis, according to the ESMO Guidelines [18]. For colorectal cancer (CRC), all level 1 genomic alterations, as classified by the ESMO Scale for Clinical Actionability of molecular Targets (ESCAT), are point mutations that can be easily detected using PCR. Therefore, ESMO recommends the use of NGS as an alternative to PCR only when it does not incur additional costs. This is because potentially actionable alterations are rare in CRC, making the systematic use of NGS not cost-effective from a large-scale perspective.

However, unlike PCR, comprehensive genomic profiling (CGP) through NGS offers significant advantages, including the ability to multiplex by analyzing multiple genes in a single test and detecting a broader range of genomic aberrations. This expanded capability can provide critical information for both clinical and research purposes, offering novel insights into patients’ prognoses and mechanisms of resistance. Additionally, the broader use of NGS could potentially facilitate increased enrollment in clinical trials, giving patients access to emerging, promising therapies.

Comprehensive genomic profiling is both feasible and cost-effective when conducted in reference centers. Therefore, greater efforts are required from local authorities to foster collaboration networks and to streamline the exchange of information and biological samples between peripheral hospitals and reference institutions. This would enable broader access to NGS platforms and enhance the clinical and research utility of genomic data in CRC management.

## Figures and Tables

**Figure 1 ijms-25-11071-f001:**
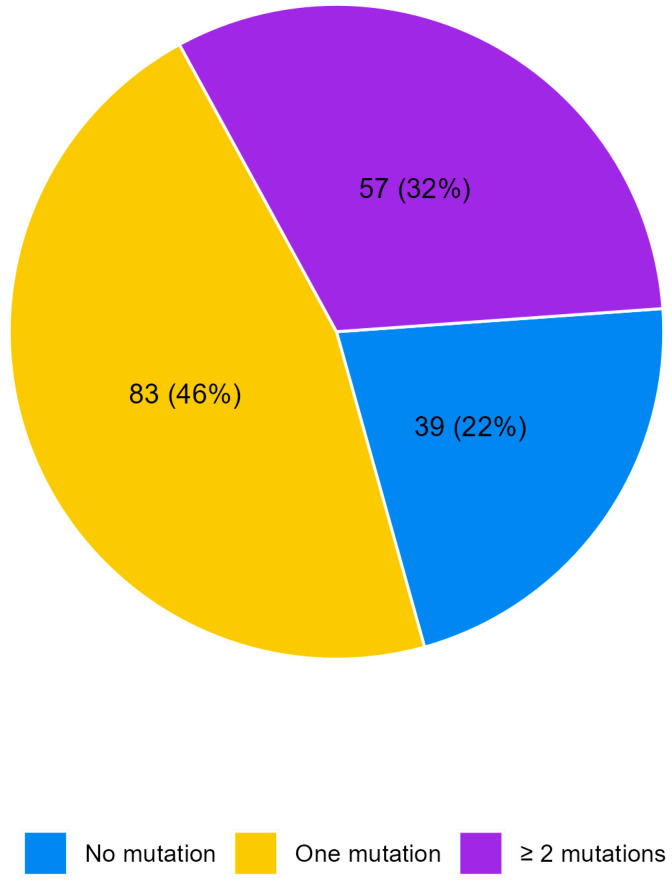
Pie chart of the number of alterations per patient.

**Figure 2 ijms-25-11071-f002:**
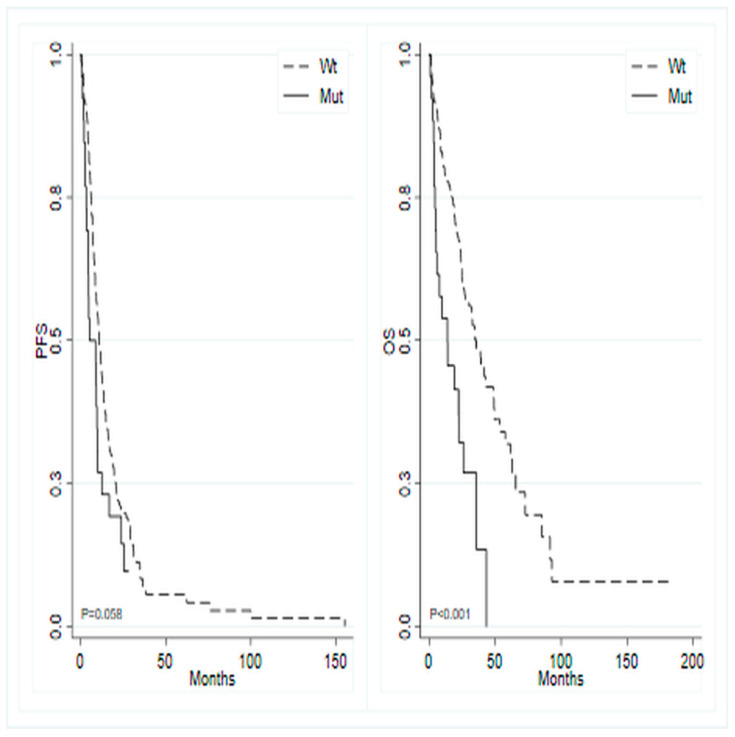
Kaplan–Meier curves for progression-free survival (**left**) and overall survival (**right**) based on the mutational status of *BRAF* gene.

**Table 1 ijms-25-11071-t001:** Patient Characteristics.

Variable	*n*	(%)
**Gender**		
Female	74	(41.3)
Male	105	(58.7)
**Median age at diagnosis (IQ-IIIQ)**	69.6 (58.3–77.4)	
**ECOG PS**		
0	109	(60.9)
1	31	(17.3)
2–3	20	(11.2)
NA	19	(10.6)
**Primary Tumor Site**		
Right Colon	64	(35.7)
Left Colon	69	(38.6)
Rectum	46	(25.7)
**Stage at Diagnosis**		
I–III	113	(63.1)
IV	66	(36.99
**Number of Metastatic Sites**		
1	115	(64.2)
>1	64	(35.8)
**Metastatic Sites**		
Liver	127	(70.9)
Lung	50	(27.9)
Lymph Nodes	34	(18.9)
Peritoneum	44	(24.5)
Bone	6	(3.3)
Pelvic Recurrence	6	(3.3)
Other	16	(8.9)
**Metastasis Surgery**		
Yes	50	(27.9)
No	129	(72.1)
**Type of Metastasis Surgery**		
Hepatic	15	(30.0)
Pulmonary	9	(18.0)
Peritoneum	10	(20.0)
Other Site	16	(32.0)
**First-Line Systemic Chemotherapy**		
Yes	149	(83.2)
No	30	(16.8)
**First-Line Chemotherapy Regimen**		
Doublet/Triplet plus bevacizumab	66	(44.1)
Doublet/Triplet plus EGFR inhibitor	37	(24.9)
Chemotherapy alone (Doublet or monotherapy)	31	(20.8)
Monotherapy + bevacizumab or EGFR inhibitor	15	(10.1)
**Number of Treatment Lines**		
1	54	(36.2)
2	40	(26.9)
3	33	(22.1)
4	16	(10.8)
≥5	6	(4.0)
**First-Line Targeted Therapy**		
Reimbursed	6	(3.3)
Clinical Trial	2	(1.1)

IQ: first quartile; IIIQ: third quartile; NA: not available; EGFR inhibitor: epidermal growth factor receptor inhibitor (cetuximab or panitumumab). Percentages may not equal 100 due to rounding.

**Table 2 ijms-25-11071-t002:** Distribution of the type of molecular alterations.

Gene	Frequency
*KRAS*	87 (48.6%)
*PIK3CA*	40 (22.4%)
*BRAF*	26 (14.5%)
*APC*	15 (8.4%)
*NRAS*	8 (4.5%)
*ERBB2*	5 (2.8%)
*SMO*	4 (2.2%)
*CTNNB1*, *MYC*, *MAP2K1*	3 (1.7%)
*ERBB3*, *RAF1*, *MTOR*, *JAK1*, *FGFR2*	2 (1.1%)
*CDK4*, *MET*, *FGFR3*, *GNA11*, *EGFR*, *ALK*, *ROS1*, *DDR2*, *KIT*	1 (0.6%)

The total does not add up to the total number of patients as the alterations are not mutually exclusive. As a secondary analysis, we observed the prevalence of actionable mutations, which was as follows: *BRAF* V600E 14.5% (*n* = 26), *PIK3CA* 22.4% (*n* = 40), *KRAS* G12C 2.8% (*n* = 5), and *MET* 0.6% (*n* = 1).

## Data Availability

The datasets generated and/or analyzed during the current study are available from the corresponding author on reasonable request.

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
