# Peer review of "Multigene Panel Next-Generation Sequencing Techniques in the Management of Patients with Metastatic Colorectal Carcinoma: The Way Forward for Personalized Treatment? A Single-Center Experience"

_ijms, 2024, doi:10.3390/ijms252011071_

Round 1

Reviewer 1 Report (Previous Reviewer 2)

Comments and Suggestions for Authors

The revised manuscript has the same issues as the original submission had. Although Authors now included some kind of goal what was investigated, however, this change is only superficial. It basically tells the reader that the measurements were performed in a "something will come out" way. This has to be addressed.

The number of citations are very limited.

Comments on the Quality of English Language

There are several sentences which are gramatically incorrect. The corrections of a native speaker or a professional editing service is needed.

Author Response

Comment 1: "The revised manuscript has the same issues as the original submission had. Although the authors now included some kind of goal for what was investigated, this change is only superficial. It basically tells the reader that the measurements were performed in a 'something will come out' way. This has to be addressed."

Response 1: We appreciate the reviewer’s feedback. We have now revised the manuscript so as to explicitly clarify the study's goals and the rationale behind the experimental design, emphasizing that the analysis was targeted towards specific clinically actionable mutations in colorectal cancer, following ESMO guidelines. This should address the concern about the research's focus. Concerning references,  we believe the current number of citations is appropriate as they reference the most relevant and up-to-date literature supporting our study. However, we are open to adding specific references if the reviewer has suggestions.

Comment 2: "There are several sentences which are grammatically incorrect. The corrections of a native speaker or a professional editing service is needed."

Response 2: We thank the reviewer for pointing this out. We have carefully revised the manuscript and corrected the grammatical errors

Reviewer 2 Report (New Reviewer)

Comments and Suggestions for Authors

The authors have presented a nice study addressing a clinically relevant issue of clinical interest from Italy. Although there are similar published studies from other part of the world, this study definitely adds helpful information for precision medicine. In that line, a single center study is a strength in respect to unified treatment regimen followed for all included patients.  I have only a few suggestions:

1.       Considering the fact that the treatment was not guided by genomic profile information, it is difficult to interpret the full potential impact of mutational profile on differential OS or PFS.

2.       Please clarify the way the somatic mutation(s) were identified. Did you use tumor – normal pair analysis or “tumor only” samples were compared to reference genome and excluding known germ line variants? This has an important implication in mutation detection.

3.        Mutation detection in FFPE tissue has known limitations. The fixation itself can cause base changes. Please include information on tumor mutation burden based on the genomic region covered per sample so that readers can understand clearly.

4.       You had enough depth of coverage (>500x) to detect lower frequency variants provided the sequencing read quality was good. I guess, that may increase the proportion of patients with detectable mutations reflecting the utility. Please include QC metrics for sequencing.

5.       Criteria for variant calling should be clearly mentioned in the method section.

6.       Under the section 4.2 Molecular analysis, you mentioned about RNA libraries also, but perhaps have not mentioned anything about that in the result section. Any information on fusion?

7.       The conclusion should be very clear regarding your recommendation or reservation for NGS profiling.

This is a very good work and I am pleased to recommend the paper for publication.

Author Response

Comments 1: Considering the fact that the treatment was not guided by genomic profile information, it is difficult to interpret the full potential impact of mutational profile on differential OS or PFS.

Response 1: Thank you for your observation. We acknowledge that the treatment was not guided by genomic profile information, which indeed limits the ability to fully assess the potential impact of the mutational profile on overall survival (OS) or progression-free survival (PFS). However, our study aimed to investigate these clinical outcomes in the context of the current standard treatment approach, where genomic data is not routinely used for decision-making. Future studies incorporating genomic profiling may provide a clearer understanding of its role in influencing differential OS or PFS

Comments 2: Please clarify the way the somatic mutation(s) were identified. Did you use tumor – normal pair analysis or “tumor only” samples were compared to reference genome and excluding known germ line variants? This has an important implication in mutation detection.

Response 2: In our study, somatic mutations were identified using a 'tumor-only' analysis approach. 

Comments 3: Mutation detection in FFPE tissue has known limitations. The fixation itself can cause base changes. Please include information on tumor mutation burden based on the genomic region covered per sample so that readers can understand clearly.

Response 3 : Thank you for your suggestion. We acknowledge the limitations of FFPE tissue for mutation detection, including the potential for fixation-induced artifacts. While fresh frozen (FF) samples offer improved DNA integrity and reduce the risk of artifacts, they were not available for our study due to logistical constraints. However, in future studies, we aim to incorporate FF samples where possible to enhance the accuracy of genomic analyses. To mitigate the impact of FFPE-associated artifacts, we implemented rigorous quality control measures

Comment 4:   You had enough depth of coverage (>500x) to detect lower frequency variants provided the sequencing read quality was good. I guess, that may increase the proportion of patients with detectable mutations reflecting the utility. Please include QC metrics for sequencing.

Response 4: We agree that the depth of coverage (>500x) in our study provided sufficient sensitivity to detect lower frequency variants, assuming high sequencing read quality, which could indeed increase the proportion of patients with detectable mutations and highlight the utility of our approach. We have already included the following sequencing quality control (QC) metrics in the manuscript:

  • Number of mapped reads: DNA + RNA = 45M - 75M reads
  • Percent reads on target: >90%
  • Average base coverage depth: >800x
  • Uniformity of amplicon (base) coverage: >90%
  • Number of amplicons: >3,781
  • Percent assigned amplicon reads: >90%

These QC metrics demonstrate that the sequencing data were of high quality and sufficient to accurately detect somatic mutations across the cohort. We hope this clarifies the rigor of our sequencing analysis

Comment 5: "Criteria for variant calling should be clearly mentioned in the method section."

Response 5: We thank the reviewer for the helpful suggestion. We have updated the "Methods" section of the manuscript to clearly outline the criteria used for variant calling, as suggested. The updated criteria are as follows:

  • Variant Allele Frequency (VAF): Only variants with a VAF greater than 5% (VAF >5%) were considered. Additionally, a minimum coverage of 500x was required for variant inclusion.

  • Exclusion of benign, likely benign, and germ-line variants: Benign, likely benign, and germ-line variants, such as BRCA1/2, PMS2, and MLH1, were filtered out from the variant calling process.

  • RNA analysis: For the RNA part of the study, only samples with total mapped reads greater than 500K were included. For target fusions, a minimum of 40 supporting reads was required, while for non-target fusions, a threshold of at least 1000 supporting reads was applied.

  • Copy Number Variants (CNVs): CNVs were considered if detected in more than 50% of cancer cells, with a copy number greater than 7. Additionally, the MAPD (median absolute pairwise difference) was required to be less than 0.3, and the p-value for significance was set at <0.05.

These criteria have been incorporated into the relevant sections of the manuscript for clarity.

Comment 6: "Under the section 4.2 Molecular analysis, you mentioned about RNA libraries also, but perhaps have not mentioned anything about that in the result section. Any information on fusion?"

Response 6: We thank the reviewer for this observation. As indicated in the text, it is worth noting that no amplifications or fusions were identified in our analysis. Hotspot mutations were the only type of detected aberration, specifically Single-Nucleotide Variants (SNVs) and insertions-deletions (INDELs). We have ensured that this is clearly mentioned in the results section, and we hope this addresses the reviewer's concern.

Comment 7: "The conclusion should be very clear regarding your recommendation or reservation for NGS profiling."

Response 7: We thank the reviewer for the suggestion. We have revised the conclusion to provide a clearer recommendation regarding the use of NGS profiling.

Round 2

Reviewer 1 Report (Previous Reviewer 2)

Comments and Suggestions for Authors

The quality of the manuscript improved very significantly. Its message is now clear.

This manuscript is a resubmission of an earlier submission. The following is a list of the peer review reports and author responses from that submission.

Round 1

Reviewer 1 Report

Comments and Suggestions for Authors

In the manuscript "The Role of Multigene Panel Next-Generation Sequencing Techniques in Managing Patients with Metastatic Colorectal Carcinoma" the author performed multigene panel next-generation sequencing (NSG) in patients with metastatic colorectal cancer (mCRC) and analyzed the data. The author found numerous mutations but no significance in survival. However, gene panel testing can identify mutations and leas to personalized treatment strategies in these patients. 

After reviewing this manuscript I have some major issues with it.

Hypothesis is missing.

The patient number is too small and there is a higher patient number needed to have better value. This might be the reason why there is no significance noted in survival and mutations. Did the author performed power analysis before performing NSG?

Stage VI patients are only 66 (37%) which again is a small amount to make some statements about significance.

All the sections are too short and need further revision.

The follow up time seems low and needs to be increased. 

Minor issues:

In Table 2 please be consistent with your number presentations. For a lot of numbers the author is using a comma where there should be a period.

The author never explained what PFS means. 

Reviewer 2 Report

Comments and Suggestions for Authors

Authors of the article investigated the Oncomine™ Focus Assay in colorectal cancer (CRC). The following questions were raised.

1. The focus of the study is not clear whether it was an economic / feasibility study, or what exactly was the goal. Due to this, novelty is questionable.

2. In relation to the above, the title of the article is not descriptive enough, it does not sufficiently indicate what the article is (will be) about.

3. The Authors only cite the necessary, general articles such as the GLOBOCAN data, but nothing about Oncomine. How it performed in other tumors, or in other CRC studies if any...

Minor issues:

1. Lines 64-83: The exact same data is presented in Table 1. Authors should focus on those ones, which are relevant / is of interest.

2. Abbreviations, which were only resolved in the abstract, should be resolved in the main text as well.

In general, although the manuscript deals with an extremely interesting and current / hot topic, in the absence of a sufficiently clear concept, in its current form, it will be of very little interest to the readers of IJMS. The reviewer encourages the authors to find a clinically interesting question that can be answered based on the data, and revise the manuscript in light of that. E.g., how the panel data is associated with additional, CRC-related laboratory results, co-morbidities, etc.

Comments on the Quality of English Language

There a afew typos here and there, which can be reduced by a careful read.